# Effect of pH on the Formation of Amorphous TiO$_2$ Complexes and TiO$_2$ Anatase during the Pyrolysis of an Aqueous TiCl$_4$ Solution

**Huyen Duong Ngoc [1,\*], Dung Mai Xuan [2]**  **and Tuan Mai Van [1,3]**

[1] School of Engineering Physics, Hanoi University of Science and Technology, No. 1, Daicoviet, Hanoi 10000, Vietnam; tuanmvns@gmail.com

[2] Department of Chemistry, Hanoi Pedagogical University 2, 32 Nguyen Van Linh, Phuc Yen, Vinh Phuc 15000, Vietnam; xdmai@hpu2.edu.vn

[3] Faculty of Natural Sciences, Electric Power University, 235 Hoang Quoc Viet, Hanoi 10000, Vietnam

[\*] Correspondence: huyen.duongngoc@hust.edu.vn; Tel.: +84-912-153-128

**Abstract:** The titanium dioxide (TiO$_2$) nanostructures resulted by the pyrolysis of titanium tetrachloride (TiCl$_4$) at a low temperature of 80 °C were found to be a mixture of amorphous TiO$_2$ complexes and anatase nanostructures, whose ratio depends on the pH of the pyrolysis medium. At a low pH level, the resulting TiO$_2$ nanostructures are predominantly anatase and gradually shift to amorphous TiO$_2$ complexes as the pH level increases. Moreover, the amorphous TiO$_2$ complexes can convert back to anatase nanostructures by a post-heating treatment, and can then transform to rutile with elevating temperature. Amongst the TiO$_2$ nanostructures recovered from the amorphous TiO$_2$ complexes, anatase appears to be the most effective photocatalyst in the decomposition of methylene blue.

**Keywords:** photocatalyst; amorphous TiO$_2$ complexes; TiO$_2$; anatase nanostructures

## 1. Introduction

Titanium dioxide (TiO$_2$), a typical metal oxide with a high refractive index, chemical stability, long durability, and nontoxicity, has been widely used for many applications, such as white pigments, textiles, papers, cosmetics, medicines, and ceramics. As an *n*-type wide bandgap semiconductor, TiO$_2$ exhibits a unique photoinduced effect involving photogenerated charge carriers that initiate a strong redox reaction of adsorbed substances and hydrophilic conversion of itself [1,2]. This effect offers more potential applications involving photochemical processes, such as splitting hydrogen from water, a photocatalyst, a photoconductor, environment cleaning, an antibacterial purpose, chemica sensors, ultraviolet filters, and dye-sensitized solar cells (DSSCs) [3–5].

Under normal conditions, TiO$_2$ exists in three main structures: stable rutile, metastable anatase, and brookite phases. For the pure phase, it is generally accepted that anatase exhibits a higher photocatalytic activity compared to that of rutile, despite its larger bandgap (3.2 eV for anatase vs. 3.0 eV for rutile). The longer lifetime for photo-excited electrons and holes in the indirect bandgap of TiO$_2$ anatase semiconductor is explained for this feature [6]. In contrast, TiO$_2$ in the microstructure is considered a poor photocatalyst, but in a nanostructured form, due to the quantum confinement, the material shows stronger photocatalytic activity in comparison to that of the microstructure [7]. The unique photocatalyst of TiO$_2$ is size- and structure-dependent. Therefore, clarification of the effect of synthesis conditions on the resulting TiO$_2$ nanostructures is of importance to yield effective photocatalysts and diverse photocatalytic applications.

With regard to the synthesis of $TiO_2$ nanostructures, a variety of techniques based on the pyrolysis of Ti precursors, such as the hydrothermal, solvothermal, sol–gel, direct oxidation, chemical vapor deposition (CVD), electrodeposition, sonochemical, and microwave methods, have been used [8]. Pyrolysis offers a simple route to synthesize well-crystalline $TiO_2$ using inexpensive precursors, such as titanium (IV) tetrachloride ($TiCl_4$), titanium (IV) butoxide, titanium (IV) isopropoxide, amorphous $TiO_2$, and $P_{25}$. In addition, the pyrolysis modest medium of low temperature and adjustable pyrolysis time can provide an effective environment for the synthesis of $TiO_2$ with high purity, good dispersion, and controllable crystalline. From the viewpoint of chemical thermodynamics, before decomposing into $TiO_2$ either in the form of anatase, brookite, or rutile, the titanium precursor undergoes a series of amorphous $TiO_2$ complexes (or intermediates) such as $Ti_xO_yCl_z(OH)_w$, resulting from the pyrolysis of $TiCl_4$, $[Ti(OH)_{4-n}(H_2O)_{2+n}]^{n+}$ from Ti(IV)-butoxide or $[Ti_{3(y+1)}O_{4y}(OBu)_{4(y+3)-x}(OEt)_x]$ from alkoxide metal $M(OR)_n$ [9–12]. Amorphous $TiO_2$ complexes in general are metastable phases, so they can evolve into any crystalline phases under special conditions. The heat treatment route with appropriate reactants is usually used to achieve the transformation from amorphous to crystalline $TiO_2$. For example, pure anatase or rutile and brookite nanoparticles are obtained by hydrothermal treatment of amorphous $TiO_2$ with a variety of acid additives as reactants at different concentrations [13–15]. Chemical vapor deposition synthesis of pure brookite $TiO_2$ thin films is also realized by using amorphous $TiO_2$ as the precursor [16,17]. Hence, the amorphous $TiO_2$ intermediates can be used as a kind of secondary precursor to produce either brookite, anatase, rutile structures, or their derivatives.

With regard to the amorphous $Ti_xO_yCl_z(OH)_w$ complexes resulted from the pyrolysis of $TiCl_4$, a calculation shows that the substitution of OH for Cl radicals in the complexes does not change much in the core involving the Ti atoms, but there is a difference in the bond lengths and potential energy surfaces [10]. From a thermodynamic point of view, a change in the relative ratio of the Cl and OH radicals modifies the potential surface energy and then the free energy of the complex $TiO_2$ intermediates, consequently affecting the final $TiO_2$ nanostructures. As examples, some works have demonstrated that the pyrolysis of aqueous $TiCl_4$ solutions with high HCl result in rutile and brookite structures [18,19]. However, anatase, which is considered to be the most active photocatalyst, was hardly observed in the resulting materials. From our perspective, the excessive Cl radicals in the reaction medium due to a large amount of HCl additive (pH < 0) resulted in the predominance of rutile and brookite. Moreover, some possible $TiO_2$ nanostructures in colloidal or amorphous forms left in the reaction solution may have been filtered and washed away. In a series of experiments made on the pyrolysis of an aqueous $TiCl_4$ solution with lesser HCl concentrations (0.0 < pH < 1.0), we found that the resulting materials were nanocrystalline mixtures of both the anatase and rutile phases. The $TiO_2$ anatase mainly suspended in the aqueous solution in the colloidal form, while the $TiO_2$ rutile predominantly precipitated and deposited in the sedimentation [20,21]. It was concluded that a high HCl concentration enabled the agglomeration of anatase particles and enhanced the anatase to rutile transition due to the compensation of a Cl radical for the positive charge of polyhedral complexes. Furthermore, the pyrolysis of $TiCl_4$ in the neutral medium, with a pH level of approximately 6–8, brought in materials of predominantly amorphous $TiO_2$, as well as trace amounts of anatase, but no rutile or brookite [14,22]. Then, the pH levels evolving to correlations between H, Cl, and OH radicals in the reaction media were assumed to be the crucial factors to modify and control the resulting $TiO_2$ nanostructures. Based on these considerations, in this study, an experiment was carried out to further investigate the effect of an elevated pH level (or OH radical) of the reaction media on the formation of amorphous $TiO_2$ complexes, as well as of the final anatase nanostructures obtained from the pyrolysis of an aqueous $TiCl_4$ solution.

## 2. Results and Discussion

The experiments show that the $NH_4OH$ additive, a weak basic agent used to adjust the pH level of the reaction solution, significantly affects the appearance and properties of $TiO_2$ nanostructures in the resulting materials. As seen in Figure 1, the resulting aqueous solution appears transparent

at a low pH level but gradually changes to slightly opalescent, and then separates into transparent and milky parts when the pH level exceeds 2.40. The separated milky column increases with $NH_4OH$ and becomes unchanged as the pH level exceeds 7.34. Depending on the transparent or milky state, the surface morphology of the resulting materials transforms from a grain to a gelation structure as shown in the Scanning Electron Microscope (SEM) images in Figure 2. In the sample with a pH of 0.98, the resulting material grains are uniformly granular with a mean size of approximately 50–70 nm, which, in turn, shows clusters of anatase nanoparticles of 4–5 nm in mean size [20]. However, when the pH level of the reaction medium increases, the resulting material grains inflate to coagulated clusters of 150–200 nm in size and gradually become jellylike or amorphous structures, as shown in Figure 2b–d.

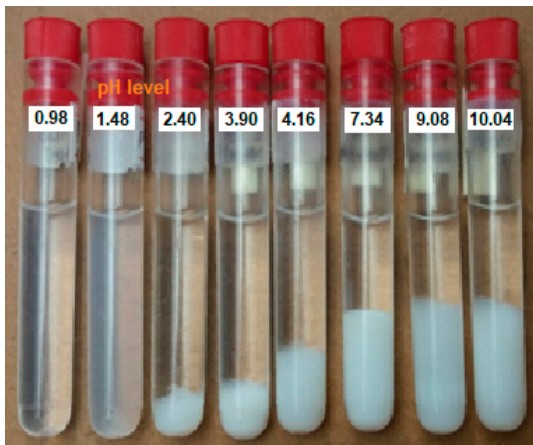

**Figure 1.** The appearance of the aqueous titanium tetrachloride ($TiCl_4$) solution with different pH levels after pyrolysis at 80 °C.

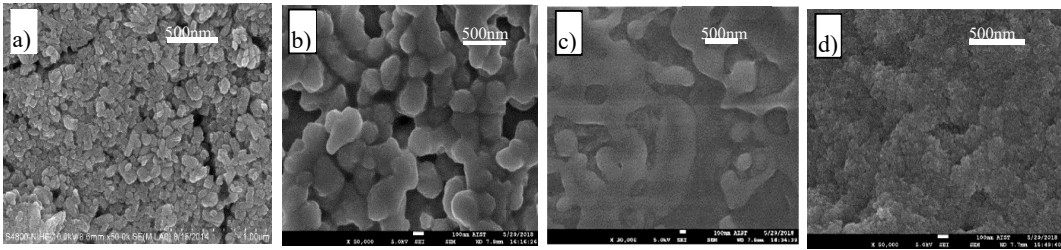

**Figure 2.** SEM images of titanium dioxide ($TiO_2$) resulting from the pyrolysis of $TiCl_4$ in different pH media at 80 °C: (**a**) pH = 0.98; (**b**) pH = 2.45; (**c**) pH = 10.04 (transparent part); (**d**) pH = 10.0 (milky part).

The X-ray diffraction (XRD) spectra in Figure 3 show the evolution of the resulting materials depending on the pH level, i.e., on $NH_4OH$ additive. In the sample with a pH level of 0.98, the XRD pattern contains a principal peak at approximately 25.29° and the other peaks at approximately 37.80°, 48.05°, 53.89°, and 62.68°, assigned to the diffractions of the anatase structure on the (101) and (004), (200), (105), and (204) planes, respectively (JCPDS no. 00-021-1272). When $NH_4OH$ was added, together with the diffraction peaks from anatase, other sharp diffraction peaks at 22.98°, 32.69°, 40.31°, 46.88°, 52.80°, 58.29°, and 68.43° were observed for the diffractions on the (100), (110), (111), (200), (210), (211), and (220) planes, respectively, with crystals emerging from $NH_4Cl$ [23]. With the increase in pH level, the $TiO_2$ diffraction in the XRD pattern gradually disappeared, accounting for the gradual conversion from $TiO_2$ anatase to amorphous $TiO_2$ complexes. Using the Scherrer equation, i.e., $D = k\lambda/\beta\cos\theta$, where $k = 0.94$, $\lambda = 0.154$ nm, and $\beta$ is full width at half maximum (FWHM) at diffraction angle $\theta$ according to (101) peak, to calculate the mean size D of the anatase particles, it was found that the mean sizes of the anatase particles were almost unchanged at approximately 4.5 nm, as given in Table 1. This value is considered to be the limitation of anatase size in the conversion to amorphous $TiO_2$ complexes.

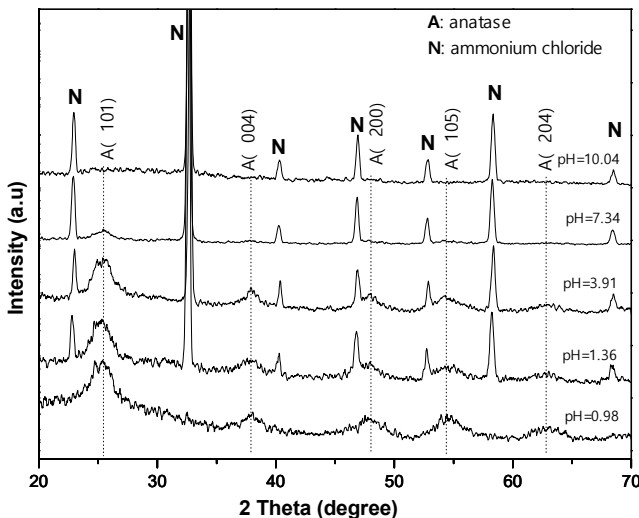

**Figure 3.** X-ray diffraction (XRD) spectra of TiO$_2$ resulting from the pyrolysis of TiCl$_4$ in different pH media at 80 °C.

**Table 1.** The mean size of the anatase particles resulting from the pyrolysis of TiCl$_4$ in different pH media at 80 °C. FWHM, full width at half maximum.

| pH | (101) Peak FWHM (°) | Size (nm) | Agent Addition |
|------|---------------------|-----------|----------------|
| 0.98 | 2.007 | 4.3 | No addition |
| 1.36 | 1.889 | 4.5 | NH$_4$OH |
| 3.91 | 1.830 | 4.7 | NH$_4$OH |
| 7.34 | 1.888 | 4.5 | NH$_4$OH |
| 10.04 | - | - | NH$_4$OH |

The Raman spectra also confirm the presence of anatase and NH$_4$Cl in the resulting materials. As shown in Figure 4, in the starting materials, namely, the sample with a pH level of 0.98, the spectrum exhibits only vibrational modes at approximately 155 cm$^{-1}$, 399 cm$^{-1}$, 513 cm$^{-1}$, and 634 cm$^{-1}$, respectively, representing the E$_g$, B$_{1g}$, A$_{1g}$ + B$_{1g}$, and E$_g$ modes of the anatase structure [24,25]. The presence of NH$_4$Cl in the materials gives rise to a broad saddle spectrum consisting of two vibration modes at approximately 168 cm$^{-1}$ and 144 cm$^{-1}$ that are assumed to be the supposition of the E$_g$ vibration mode of anatase and the $\nu_2$, $\nu_3$, and $\nu_4$ vibration modes of NH$_4$Cl oscillating against Cl along the (100) direction and along three orthogonal directions [26].

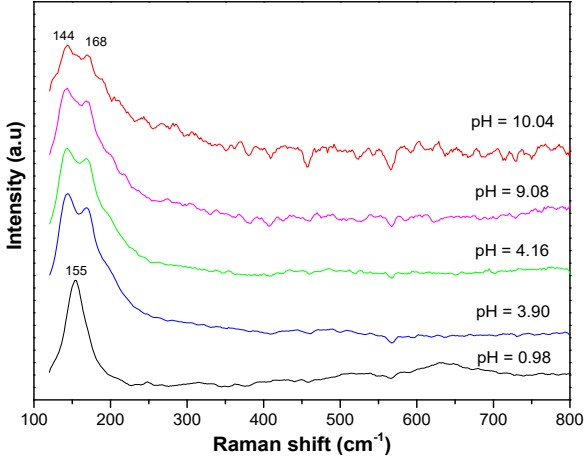

**Figure 4.** Raman spectra of TiO$_2$ resulting from the pyrolysis of TiCl$_4$ in different pH media at 80 °C.

The appearance of transparent, opalescent, and separable milky parts in the solution is believed to be due to the appearance and increase of amorphous $TiO_2$ complexes forming in the pyrolysis medium. At a low pH level, in the acidic aqueous medium with a higher concentration of H and Cl radicals, the formation of amorphous $TiO_2$ complexes is negligible; the resulting anatase is crystallized in the form of grain structures with sharp boundaries. The presence of $NH_4OH$ in the pyrolysis medium raises the pH level and then the OH radicals that promote the formation of amorphous $TiO_2$ complexes. Consequently, with the increase in $NH_4OH$ additive, the separated milky fraction in the medium is gradually increased, in agreement with the gradual decrease of anatase diffraction in the XRD spectra. When the pH level exceeds 7.34, the milky column is unchanged, even though the $NH_4OH$ additive keeps increasing. Furthermore, the Energy Dispersive X-ray Spectra (EDS) show that no trace of Ti is present in the transparent part but is in the milky part, as shown in Figure 5. This indirectly indicates that the decomposed $TiCl_4$ precursor in the pyrolysis solution was totally converted into amorphous $TiO_2$ complexes and completely separated into the milky part as the pH level exceeded 7.34. Due to the amorphous nature, no crystalline diffraction pattern can be observed in XRD spectra as the materials were synthesized in the medium with the pH level beyond that point.

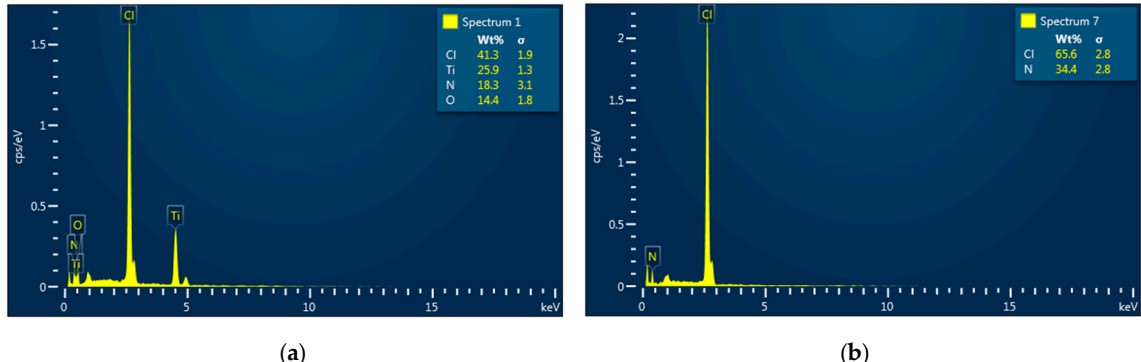

(**a**)        (**b**)

**Figure 5.** EDS spectra of the sample extracted from the milky part (**a**) and from the transparent part (**b**), extracted from the pyrolysis of $TiCl_4$ in a medium with pH = 10.04.

A High-resolution Transmission Electron Microscope (HRTEM) image taken from a milky sample with a pH level of 10.04, as given in Figure 6, shows the appearance of tiny nanocrystallites scatteringly embedded in an amorphous medium. The amorphous medium surrounding the materials is considered to be the $TiO_2$ complexes. The lattice spacing of the tiny nanocrystallites at approximately 0.346 nm is identical to the lattice spacing of the (101) plane of $TiO_2$ anatase. The estimated size of the $TiO_2$ anatase particles is comparable to that calculated from the XRD pattern, approximately 4.5 nm, which is considered to be the size limitation of anatase in equilibrium with amorphous $TiO_2$ complexes. The presence of anatase nanoparticles embedded in the amorphous $TiO_2$ complexes elucidates the appearance of the $E_g$ vibration mode of anatase in the Raman spectra in Figure 4.

In order to further characterize the evolution of the amorphous $TiO_2$ complexes into the other crystalline forms, a post-heating treatment was made at an elevated temperature of up to 600 °C. The experiments reveal that the materials underwent a conversion back to anatase and then from anatase to rutile. At a heating temperature below 200 °C, the XRD pattern in Figure 7 shows only the presence of $NH_4Cl$ but no trace of $TiO_2$ structures. However, when the heating temperature exceeds 200 °C, anatase diffraction gradually emerges while $NH_4Cl$ diffraction gradually disappears in the XRD patterns. The disappearance of $NH_4Cl$ is accounted for by the decomposition of the materials, while the appearance of $TiO_2$ anatase explains the decomposition and recrystallization of amorphous $TiO_2$ complexes at an elevated temperature. When the heating temperature exceeds 300 °C, the $NH_4Cl$ is completely decomposed and the complexes are totally converted into $TiO_2$ nanostructures that are predominantly anatase. Brookite and rutile structures are hardly observed in the XRD patterns and can be neglected in the conversion process. When the heating temperature exceeds 450 °C, the appearance

of rutile diffraction in the XRD patterns indicates the onset of the anatase–rutile transition. The mean size of anatase calculated from the XRD patterns was found to grow from approximately 4.5 nm at a heating temperature of 200 °C to 8.9 nm at a heating temperature of 600 °C, as given in Table 2. For the sample with a pH level of 0.98, anatase is predominant over amorphous $TiO_2$ complexes; the heat treatment is merely a means to enable the separation of anatase nanoparticles from the cluster.

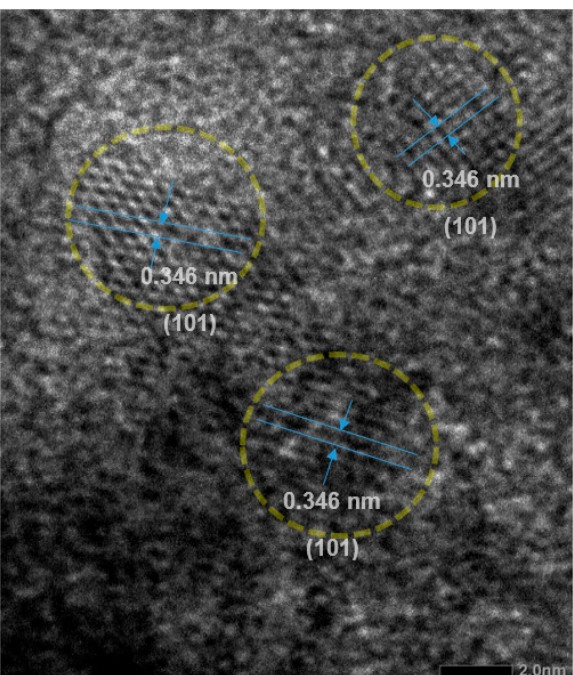

**Figure 6.** The appearance of anatase nanoparticles scatteringly embedded in the amorphous $TiO_2$ complexes.

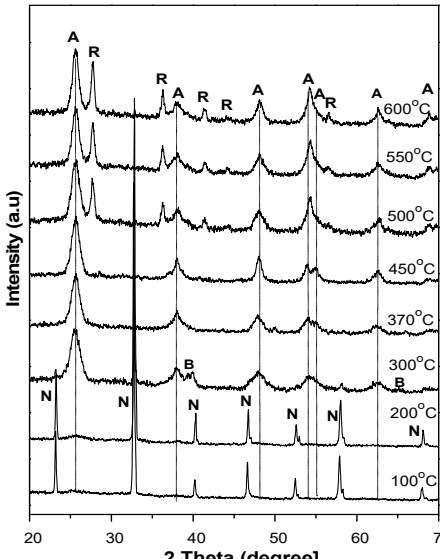

**Figure 7.** XRD spectra of $TiO_2$ resulting from the pyrolysis of $TiCl_4$ in a medium with pH = 10.04 at 80 °C and then treated at different elevated temperatures. A, anatase; B, brookite; R, rutile; N, ammonium chloride.

**Table 2.** The mean size and $E_g$ vibration mode of anatase resulting from annealing amorphous $TiO_2$ complexes.

| Baking Temp. | Crystallite Size (nm) | $E_g$ Mode Peak ($cm^{-1}$) | FWHM of $E_g$ Mode ($cm^{-1}$) |
|---|---|---|---|
| 200 °C | 4.5 | 154 | 31 |
| 300 °C | 6.2 | 150 | 25.8 |
| 450 °C | 6.8 | 148 | 18.9 |
| 500 °C | 6.8 | 148 | 18.1 |
| 550 °C | 8.3 | 147 | 15.9 |
| 600 °C | 8.9 | 145 | 14.5 |

The Raman spectra also verify the conversion of anatase from the amorphous $TiO_2$ complexes (the milky part) when heated up, as given in Figure 8. With a heating temperature below 200 °C, the Raman spectrum is a composition of the $NH_4Cl$ vibration mode centered at approximately 168 $cm^{-1}$ and 144 $cm^{-1}$ and the $E_g$ vibration mode of $TiO_2$ anatase at 147 $cm^{-1}$. As the heating temperature increases from 200 to 600 °C, the $E_g$ vibration mode shows a shift in frequency from 154 to 145 $cm^{-1}$ and a shrinkage in FWHM (see Table 2). This feature accounts for the size growth from 4.5 to 8.9 nm of the $TiO_2$ anatase nanocrystallites [27].

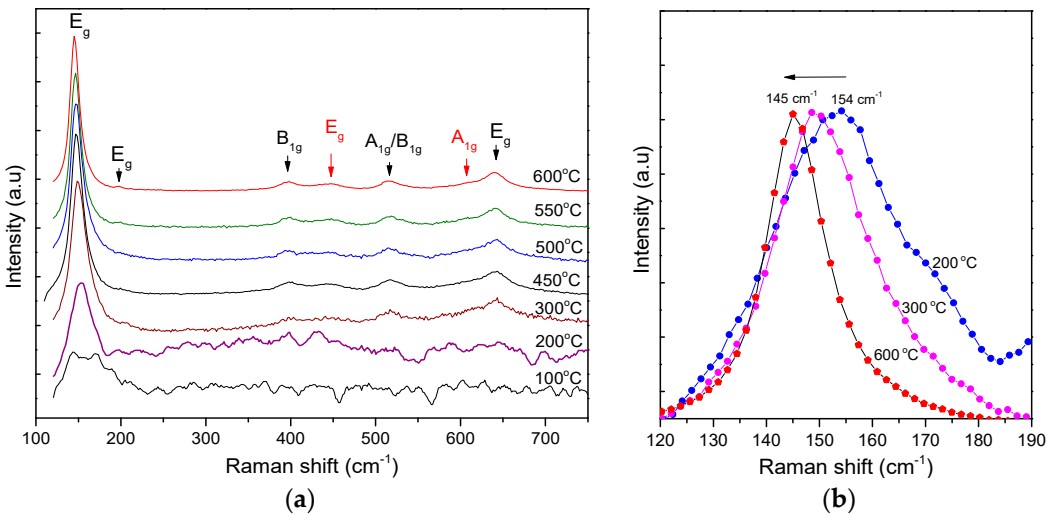

**Figure 8.** (**a**) Raman spectra of $TiO_2$ resulting from the pyrolysis of $TiCl_4$ in a medium with pH = 10.04 at 80 °C and then treated at different temperatures; (**b**) blue shift of the $E_g$ vibration mode as the heating temperature increases from 200 to 600 °C.

The formation of $TiO_2$ nanostructures by the pyrolysis of $TiCl_4$ in an elevated pH medium can be explained by two mechanisms; one is the decomposition, dissolution, and recrystallization and the other is the in-situ transition [20,28]. At an elevated temperature exceeding 80 °C, $TiCl_4$ is decomposed and hydrolyzed into HCl and amorphous $Ti_xO_yCl_z$ (or $Ti_xO_yCl_z(OH)_w$) complexes and then converted into $TiO_2$ anatase nanostructures [10]. The component ratio of the amorphous $TiO_2$ complexes and $TiO_2$ anatase is established by an equilibrium balance between the H, OH, Cl, and $NH_3$ radical concentrations in the medium. At a low pH level of 0.98 to 2.40, excessive H and Cl radicals promote the formation of anatase nanocrystallites, which likely follows the first mechanism. In this medium, the mean size of the anatase particles is approximately 4.5 nm below the limitation for the anatase to rutile transition [29–31]. Consequently, no rutile trace can be observed in the XRD diffraction pattern. In contrast, at a high pH level, the presence of OH and $NH_3$ radicals reduces and eliminates the activity of the Cl radical and brings in OH-dominant amorphous $Ti_xO_y(OH)_w$ or $[Ti(OH)_{4-n}(H_2O)_{2+n}]^{n+}$ complexes with a consumption of the anatase. As a result, at a low pH level (−1.0 to 2.40), the anatase fraction is dominant, while the amorphous $TiO_2$ complexes are dominant at a high pH level (>7.34). The amorphous $TiO_2$ complexes can be converted back to $TiO_2$ anatase nanoparticles by a post-heat treatment at a temperature of

approximately 300 °C. The amorphous–anatase transition here likely follows the in-situ transition mechanism. The relatively low transition temperature (approximately 300 °C) from amorphous $TiO_2$ to anatase nanoparticles is explained for the nucleation seeding effect that is due to the embedded anatase nanoparticles in the amorphous $TiO_2$ medium [14].

The experiments show that the amorphous $TiO_2$ complexes exhibit strong photocatalytic activity upon exposure to UV light radiation. By comparison of the relative intensity of the methylene blue (MB) principal adsorption peak in the UV–Vis spectrum, the percentage of oxidated MB in the solution was deduced and then the photocatalytic activity of the materials was calculated.

With regard to the photocatalysis of amorphous $TiO_2$ complexes, the rates of photocatalytic oxidation of MB show an exponential reduction that is well fitted to the Langmuir–Hinshelwood (L–H) kinetics model [32]. As illuminated in Figure 9a, when the MB concentration is small, the L–H equation can be simplified to an apparent first-order: $\ln(C_0/C_t) = kt$ or $C_t = C_0 \exp(-kt)$, where $C_0$ is the initial concentration of MB, $C_t$ is the concentration of the MB at UV illumination time t, and k is a constant standing for the photocatalytic redox or reaction rate. By fitting the MB decomposition curve (in Figure 9a) to the L–H equation, the dependence of the photocatalysis of amorphous $TiO_2$ complexes in terms of reaction rate on the heating temperature can be observed (Figure 9b). At a heating temperature bellow 200 °C, the photocatalysis (reaction rate) of the materials is weak but rapidly increases with increasing temperature, then reaches the maximum reaction rate at a heating temperature of approximately 300 °C. By further increasing the heating temperature, the photocatalysis of the materials declines. When $NH_4Cl$ is evaporated and eliminated, the appearance of anatase nanostructures due to the decomposition and recrystallization of amorphous $TiO_2$ complexes in the materials accounts for such behavior. At a heating temperature of approximately 300 °C, the $NH_4Cl$ and amorphous $TiO_2$ complexes are assumed to be totally decomposed, and the materials completely convert into pure anatase and exhibit the maximum photocatalysis. By further increasing the heat temperature, the photocatalysis is reduced due to the growth of anatase particles and the appearance of a rutile fraction resulting from the anatase–rutile transition.

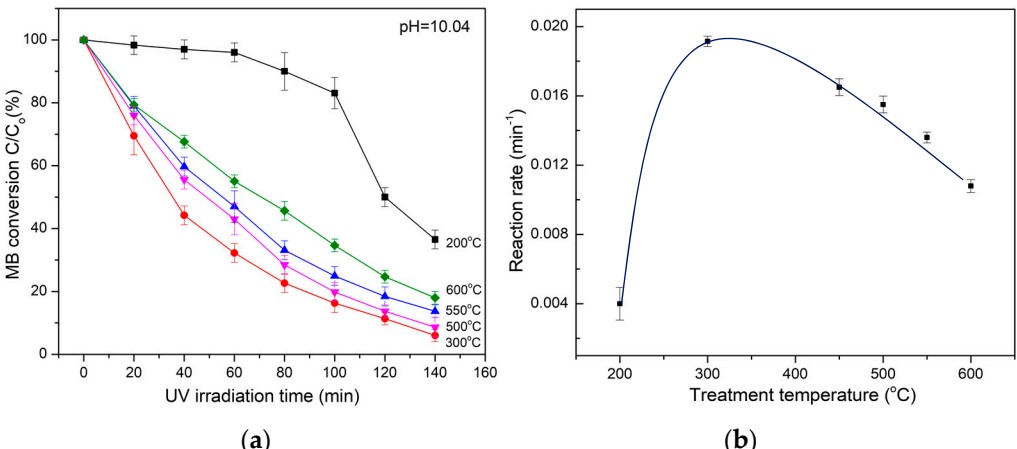

**Figure 9.** (**a**) Rates of photocatalytic oxidation of methylene blue (MB) as a function of $TiO_2$ complexes treated at different heating temperatures; (**b**) the photocatalytic activity of amorphous $TiO_2$ complexes depends on the heating temperature.

## 3. Materials and Methods

Titanium tetrachloride ($TiCl_4$) of 99.9% purity (Sigma-Aldrich Chemical Co., USA) as Ti precursor was used as received. An ammonium hydroxide solution ($NH_4OH$) of 28% $NH_3$ (Merck Corp., USA) was used as a basic agent to adjust the pH of the pyrolysis medium. The $TiO_2$ synthesis process was straightforward, as follows: $TiCl_4$ was added dropwise into deionized water at 5 °C to a concentration of 0.04 M. The pH level of the solution was then dropped to approximately 0.98 (starting point).

By adding a small amount of $NH_4OH$ into the solution, the pH level was adjusted and preserved at a point in the range of 0.98 to 10.04. The solution was then poured into test tubes and placed in an oven at 80 °C, the onset point of $TiCl_4$ decomposition. The solution was gradually changed to an opalescent suspension, indicating that the $TiCl_4$ was thermally decomposed and converted into $TiO_2$ complexes, and then $TiO_2$ nanostructures accompanied the formation of HCl and $NH_4Cl$ [12]. The pyrolysis was carried out for approximately 3 h; then, the power was shut down and the solution was slowly cooled to room temperature. Depending on the pH level, the appearance of the resulting solution showed either transparent, opalescent, or a clear split into transparent and milky parts, as seen in Figure 1. For characterization, these parts were separated and dried by vacuum evaporation, then thermally post-treated in an oven with a heating temperature up to 600 °C.

The structure of the resulting materials was determined by a D8 Advance Bruker diffractometer using $CuK_\alpha$ radiation of a 0.154 nm wavelength. The mean size, D, of the $TiO_2$ crystallites was calculated using the Scherrer equation, i.e., $D = k\lambda/\beta cos\theta$, where $k = 0.94$, $\lambda = 0.154$ nm, and $\beta$ is full width at half maximum (FWHM) according to the principal diffracted angle $\theta$, i.e., (101) peak for anatase. The Raman spectra were obtained on a LabRAM HR800 (Horiba) using a 632.8 nm excitation laser at a resolution of $1.0$ cm$^{-1}$. High-resolution Transmission Electron Microscope (HRTEM) images were obtained using a JEOL JEM-2100 TEM. Scanning Electron Microscope (SEM) images were conducted on a JEOL JEM-7600F Field Emission SEM. The photocatalytic activity of the $TiO_2$ nanostructures was determined by measuring the degradation rate of methylene blue (MB) under UV light radiation. Normally, a mixture of 50 mL of 0.25 μmol MB aqueous solution and 50 mg of amorphous $TiO_2$ complexes was stirred magnetically under dark conditions for 30 min before being exposed under a UV mercury vapor lamp. After a fixed UV exposure duration, 1 mL of the aqueous solution was taken out for UV–Vis characterization. In a diluted MB solution, the MB absorbance in the UV–Vis spectra is linearly proportional to the MB concentration according to the Lambert–Beer law. The degradation rate of MB in the solution under UV light radiation was then deduced by comparing the intensity of the MB absorbance at the maximum absorption peak of 661 nm in the UV–Vis spectrum. The experiment was replicated several times with different samples under different exposure doses to obtain the relevant results. The UV–Vis was carried out using a Cary 100 UV–Visible Spectrophotometer (Agilent).

## 4. Conclusions

The pyrolysis of an aqueous $TiCl_4$ solution generally results in a mixture of anatase nanostructures and amorphous $TiO_2$ complexes. The ratio of $TiO_2$ anatase nanostructures to amorphous $TiO_2$ complexes can be controlled by changing the pH of the pyrolysis medium. The anatase fraction is predominant at a low pH level and gradually declines and completely converts to the amorphous $TiO_2$ complexes at a high pH level. The pyrolysis of a 0.04 M aqueous $TiCl_4$ solution brings about a mixture of $TiO_2$ anatase nanostructures and amorphous $TiO_2$ complexes at a pH below 7.34 and yields predominantly amorphous $TiO_2$ beyond that point.

The amorphous $TiO_2$ complexes are found to be converted to $TiO_2$ nanostructures by heat treatment. With an annealing temperature of approximately 300 °C, the amorphous $TiO_2$ is completely converted into anatase nanostructures and gradually transforms into rutile at a high temperature. Amongst the $TiO_2$ nanostructures recovered from the amorphous $TiO_2$ complexes, the anatase nanostructure is the most effective photocatalyst in the decomposition of methylene blue.

**Author Contributions:** Conceptualization, H.D.N. and T.M.V.; Methodology, H.D.N.; Formal Analysis, H.D.N., D.M.X. and T.M.V.; Investigation, H.D.N. and T.M.V.; Resource, H.D.N. and T.M.V.; Writing-Original Draft Preparation, H.D.N. and T.M.V.; Writing-Review and Editing, H.D.N. and D.M.X.; Supervision, H.D.N. and D.M.X. All authors have read and agreed to the published version of the manuscript.

**Funding:** The authors gratefully acknowledge the financial support received in the form of a Basic Research Project Grant in Aid (T2008-PC-123) provided by Hanoi University of Science and Technology (HUST), Vietnam.

**Conflicts of Interest:** The authors declare no conflict of interest.

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
