# Peer review of "Effect of pH on the Formation of Amorphous TiO2 Complexes and TiO2 Anatase during the Pyrolysis of an Aqueous TiCl4 Solution"

_catalysts, doi:10.3390/catal10101187_

Round 1

Reviewer 1 Report

This re-submission deals with some of the previously raised problems. However, several issues remain:

  • The writing still contains far too many errors, typos, etc. Whilst perfect standard english is not expected, there is the requirement that the writing be clear and intelligible. This is not the case in many parts of the manuscript. Furthermore, basic spelling and construction errors should not be present at all!
  • While attempts have been made to clarify and expand on the reasoning and justification for the study, I think the hypothesis still lacks clarity. Likewise, the reason for the chosen methodology/experimental approach. There needs to be a full justification of every decision made. Please refine the introduction and clearly highlight this critical aspect. 
  • I am a little confused by the process as described for the XRD data and the changes observed in response to heating. The study pyrolises the precursor. And them anneals the precursor. If that is the case, what is the function of the pyrolysis step?
  • Is it possible to add error bars to the values? It would give more confidence in the validity of the data as well as allowing the reader t reflect on the goodness of fits, etc.

Author Response

Dear sir,

Thank you for the comments. I have sent the response as attachment. Please see the attachment.

Best

Reviewer 2 Report

The paper deals with pH effect on the formation of amorphous TiO2 complexes and TiO2 anatase during TiCl4 pyrolysis. The subject of the manuscript can be of interest for the scientific community but it should not be published before several issues are addressed.

  1. English language in the manuscript is not acceptable and needs important improvements (e.g., many small mistakes like typos).
  2. The scientific novelty of the paper is not enough expressed. What is the real significance of performed research for the development of new photocatalysts? It is rather obvious that anatase form of titania has the highest photocatalytic activity in the decomposition of organic compounds.
  3. The number of references should be higher to extend the discussion part.
  4. The amorphous content should be determined using e.g. NiO as an internal standard, see the paper of Ohtani et al., J. Photochem. Photobiol. A:Chem. 2010, 216, 179.

Author Response

Dear Sir,

Thank you for the comments. I have read and response as attachment. Please see the attachment.

Best

Reviewer 3 Report

Evaluation

Effect of pH on the formation of amorphous TiO2

complexes and TiO2 anatase during pyrolysis of

aqueous TiCl4 solution

Comments:

  1. The aim of this paper was to examine the effect of pH on the pyrolysis of TiCl4 which results TiO2 nanostructures consisting anatase, rutile and brookite in different ratios. This topic could be considered as a very significant and up-to-date research area. However, after going through the publications in this field, three well-written articles about the effect of hydrolysis conditions of TiCl4 were found from which only the first one was cited (a).

  1. a) Lee, J. H.; Yang, Y. S. Journal of the European Ceramic Society 2005, 25, 3573–3578.

Its title: ‘Effect of HCl concentration and reaction time on the change in the crystalline state of TiO2 prepared from aqueous TiCl4 solution by precipitation’

  1. b) Lee, J. H.; Yang, Y. S. Materials Chemistry and Physics 2005, 93, 237–242.

Its title: ‘Effect of hydrolysis conditions on morphology and phase content in the crystalline TiO2 nanoparticles synthesized from aqueous TiCl4 solution by precipitation’

  1. c) Lee, J. H.; Yang, Y. S. Korean Ind. Eng. Chem. 2007, 18, 545–551.

Its title: ‘Synthesis of Pure Brookite-type TiO2 Nanoparticles from Aqueous TiCl4 Solution with controlled Acidity by Precipitation Method’

Authors should have highlighted the differences from these papers. As the effect of pH was examined by Lee, J. H.; Yang, Y. S., this paper does not contain enough novelty that reaches the level of Catalysts.

  1. Showing the activity of the resulted TiO2 nanoparticles, the authors examined that in the photocatalytic oxidation of methylene blue (MB). As in more publications (d, e), MB was photocatalytically degraded in a shorter period with ZnO or TiO2-MoS2 with higher ratio, these experiments showed that the TiO2 nanoparticles are much less effective.
  2. d) Harish, S.; Navaneethan, M.; Archana, J.; Silambarasan, A.; Ponnusamy, S.; Muthamizhchelvana, C.; Hayakawa, Y. Dalton Transactions 2015, 44, 10490–10498.

Its title: ‘Controlled synthesis of organic ligand passivated ZnO nanostructures and their photocatalytic activity under visible light irradiation’

  1. e) Ibukun, O; Evans, P. E.; Dowben, P. A.; Jeong, H. K. Chemical Physics 2019, 525, 110419.

Its title: ‘Titanium dioxide-molybdenum disulfide for photocatalytic degradation of

methylene blue’

  1. It has to be noted, that this manuscript is full of typographic errors. For instance, Referee found 19 major typos:

ROW

TYPO

1.

42

photocatalystic

2.

96

asigning

3.

100

increaing

4.

101

desapeared

5.

116

asumed

6.

117

oscilating

7.

127

anh

8.

133

precursor

9.

134

armophous

10.

134

partten

11.

152

demontrate

12.

157

disapearance

13.

162

negleted

14.

164

ruttile

15.

184

radiacal

16.

211

photoctalitic

17.

219

asumed

18.

247

photocatalystic

19.

247

nanostrutures

Furthermore, the author did not use subscript in cases when it should have been necessary, instead changed the font size. The References part is not edited at all.

  1. All things considered, this manuscript addresses an important research topic, however this field was published in 2005 and in 2007 in three relevant articles. Finally, extensive editing of English language and style would be required. Therefore, I suggest the rejection of this article in Catalysts.

Author Response

Dear Sir,

Thank you for the useful comments. I have read and response as attached file. Please see the attachment.

Best

Round 2

Reviewer 1 Report

While I appreciate the efforts made to amend the manuscript, I am a little alarmed by the approach taken to error bars (if I have read and understood the data and explanation correctly). Errors are not a cosmetic addition, but an intrinsic part of data analysis. They allow for data validity and significance to be determined. As far as I understand it, the errors added are inappropriate. The data has not been properly analysed and the relevance of the findings has not been justified. This is a significant oversight. 

Author Response

Dear Sir,

Thank you for the response. We highly appreciate your helpful comments and notations, however, we don’t think that we have used intrinsic part of data analysis
and allowed data validity. In order to explain in more detail our analysis and calculation and ask you a revision we enclose in this message  "Response to Reviewer Comment" for your reference.

Best regards 

Duong Ngoc Huyen

Reviewer 2 Report

The manuscript is acceptable in the present form.

Author Response

Dear Sir,

Thank you for the acceptance. We highly appreciate your comments and supports. We also have revised the manuscript and would like to ask you to commend the newer version to us again.

Best Regards

Duong Ngoc Huyen

Reviewer 3 Report

Evaluation of the resubmitted manuscript after rejection

In the revised manuscript the Authors addressed the major and minor issues raised in the original manuscript.  They have responded to all of the remarks that made me question the publication worthiness of their article. The authors have made relatively major additions in their manuscript. Hence, my recommendation has been changed compared to the initial review. Therefore, I suggest to accept of this manuscript.

Author Response

Dear Sir,

Thank you for the acceptance. We highly appreciate your comments and supports. We also have revised the manuscript and would like to ask you to commend the newer version to us again.

Best Regards

Duong Ngoc Huyen

This manuscript is a resubmission of an earlier submission. The following is a list of the peer review reports and author responses from that submission.

Round 1

Reviewer 1 Report

OVERVIEW: This manuscript refers to the pH-dependent formation of mixed phase titanium-oxygen systems. The products were subsequently tested against methylene blue for photocatalytic degradation capability. The paper requires certain material and aesthetic changes as well as additional clarifying information before it can be considered for publication.

LANGUAGE: Generally, the language is fine, although typos and errors abound. E.g.;

  • Pg 4, line 104: presence of NH4Cl in the …?
  • Pg 7, line 120: rutile, etc.

FIGURES: Figure 9 is very confusingly constructed. I would recommend a clearer representation, or split the data into separate graphs.

METHODOLOGY:

  • The methodology is improperly placed in the manuscript. Please follow the standard layout.
  • What was the preparation method for the SEM samples? Be specific.
  • What were the XRD acquisition parameters?
  • What model/fit was used for the Raman band fitting?

REFERENCES: An insufficient number (only 17!) of references have been cited. The TiO2-based literature is vast, and there are many relevant and overlapping pieces that would be appropriate for this manuscript to draw attention to (and compare against), both in the introduction as well as in the main body of the results. For example:

  • DOI: 10.1038/s41598-017-15364-y
  • DOI: 10.1016/j.jeurceramsoc.2004.09.024
  • DOI: 10.1021/acs.chemmater.7b04944
  • DOI: 10.1021/jp102020h
  • DOI: 10.1023/B:JSST.0000016134.19752.b4
  • DOI: 10.1021/jp3079887

Amongst a whole host of other, relevant literature sources that have reported on similar, preceding work.

Reviewer 2 Report

In this study, NH4OH was added to a HCl medium to control the pH level and the TiO2 nanostructure was prepared by pyrolysis of TiCl4. The structural change of titanium dioxide was reported when the pH of the solvent changes. I think the experiment itself is systematically carried out carefully. However, I think that more careful consideration should be given to NH4Cl generated by adding NH4OH for pH adjustment. In addition, Also, it is known that the anatase nanostructure exhibits strong catalytic activity, and this paper is unfortunately lacking in novelty. So, I do not recommend that the paper is published in Catalysts in the current form.

Some comments are attached below.

  • Enter the scale in Figure 2.
  • The diffraction peaks in Figure 3 should be written as 101, not (101).
  • Is PH=0.93 in Figure 3 wrong with 0.98? Such a typo makes the research less compelling.
  • There are many typographical errors. For example, "unchnged" on line 123, "arouns" on line 153, and "beond" on line 204. It would be helpful if you could proofread English properly before submitting.